# Virtual Accessible Bilingual Conference Planning: The Parks Accessibility Conference

**DOI:** 10.3390/ijerph20032302

**Published:** 2023-01-28

**Authors:** Alison Whiting, Yashoda Sharma, Manjit Grewal, Zeyad Ghulam, Waqas Sajid, Neha Dewan, Melanie Peladeau-Pigeon, Tilak Dutta

**Affiliations:** 1KITE Research Institute, Toronto Rehabilitation Institute, University Health Network, Toronto, ON M5G 2A2, Canada; 2Institute of Biomedical Engineering, University of Toronto, Toronto, ON M5S 3G9, Canada

**Keywords:** disabilities, virtual conference, Zoom Webinar, accessible communication, national parks, Parks Accessibility

## Abstract

Virtual events have become more popular recently, and while these events have the potential to be inclusive to a broader range of attendees, there is limited information available on how to plan and deliver a virtual, accessible, and bilingual event. The objective of this paper is to share how our team planned and delivered a virtual conference that was fully bilingual and accessible to individuals with disabilities by incorporating closed captions, sign language interpretation, language interpretation (audio), regularly scheduled breaks, and a multi-sensory experience. We describe our approaches to planning the conference, such as including individuals with disabilities in decision-making, selecting virtual conference platforms, captioners, and interpreters, and how we incorporated a multi-sensory experience. The paper also summarizes feedback we received from our attendees using a post-conference evaluation survey and our team’s reflections on positive aspects of the conference and opportunities for improvement. We conclude by providing a set of practical recommendations that we feel may be helpful to others planning virtual accessible bilingual conferences in the future.

## 1. Background

The COVID-19 pandemic has led to an increase in the popularity of virtual conferences and meetings that are being planned [1,2]. These virtual conferences have many advantages over in-person events including being less time-consuming, more affordable, and more inclusive for people who have been largely underrepresented in the past when most conferences were held in-person. In particular, these online events make it easier for individuals with disabilities, as well as people who provide care to individuals with disabilities, to participate [3].

However, many virtual conferences have been found to present barriers to individuals with disabilities [4,5,6]. For instance, most virtual conferences and meetings do not include captioning or sign-language interpretation, and do not ensure documents are compatible with screen readers [3]. In countries such as Canada, that have two official languages, a growing focus on improving accessibility has led to a growing need for accessible conferences that can also be delivered bilingually (in English and French) [7]. Our team was funded to make recommendations to improve the accessibility of Canada’s national parks for individuals with disabilities. Part of this project’s scope included running a bilingual virtual conference with stakeholders to better understand existing barriers to accessibility and potential accommodations that could be implemented to overcome these barriers.

The United Nations’ Convention on the Rights of Persons with Disabilities (CRPD) provides broad guidelines on access and equity issues for persons with disabilities, including information pertaining to accessing information. The CRPD, Article 21, notes that persons with disabilities have the right to seek information “through all forms of communication of their choice”, including accessible technology, availability of sign language, augmentative and alternative communication [8]. Conferences are a well-known venue for knowledge and information sharing and seeking, both in person and virtually using online technologies. While our team found some existing guidelines for running accessible conferences, as well as best practices for virtual conferences in general, we were unable to find guidance on hosting a virtual, accessible, and bilingual conference [9,10,11]. We also contacted several virtual event production companies that claimed to offer accessible virtual conference platforms. However, upon closer inspection, we discovered none of these platforms would be able to meet our needs. This led our team to develop our own approach, which is described below.

## 2. Objective

The objective of this paper is to share the approach we took, along with the key lessons we learned in planning and delivering our Parks Accessibility Conference in the summer of 2022, as a case study.

We include a breakdown of the approximate costs of each of the vendors we engaged, the roles and responsibilities of each member of our conference team, as well as opportunities to improve on our approach. Our hope is that our experience will help others with their own accessible virtual conference planning and further the opportunities for events in the virtual space that are engaging, accessible, and inclusive.

In the remainder of the paper, we summarize the objectives of our Parks Accessibility Conference (Section 3), discuss our planning approach (Section 4), describe our process for selecting the vendors for all goods and services (Section 5), and provide details on how the final conference was delivered (Section 6). We then share the results of our post-conference evaluation survey and our team’s reflections on the conference (Section 7), and conclude with a summary of recommendations for future virtual, bilingual, accessible conferences (Section 8).

## 3. National Park Accessibility

The Parks Accessibility Conference 2022 was part of a 3-year research project, Improving Accessibility of our National Parks, that was funded by Accessibility Standards Canada in October 2019 (research ongoing). The objective of our project was to study the barriers that individuals with disabilities face when accessing Canada’s national parks and make recommendations to improve access and remove barriers.

We wanted our conference to bring together members of the disability community, organizations that support these individuals, and researchers to share their perspectives on how we can make our national parks more accessible. The conference took place from 12 p.m. to 4 p.m. (EDT) each day over three days (August 23–25). Each day focused on a different theme:Day 1: What does park accessibility mean to you?Day 2: What are barriers to national park accessibility?Day 3: What are innovative solutions that can improve park accessibility?

The primary aim of the conference was to make it as accessible as possible for individuals with disabilities to ensure we included the broadest possible range of potential park visitors. In particular, we wanted to ensure there were mechanisms to achieve the following:Make visual elements accessible to attendees with visual impairments;Make audio elements accessible to attendees who were D/deaf or hard of hearing;Avoid overstimulation for individuals with cognitive impairment;Create ways to incorporate multi-sensory experiences remotely.

Secondary aims of the conference were as follows:Deliver the conference in both English and French;Schedule the event such that it would be reasonably convenient for attendees across all North American time zones to join live;Create a networking opportunity for people that have a shared interest in improving park accessibility;Record all conference presentations to make it possible for viewing at a later time;Allow partial participation by individuals who only had access to a conventional (landline) telephone (no access to a computer or mobile device).

## 4. Conference Planning

### 4.1. Theoretical Frameworks: Co-Design Framework and Inclusive Design Theory

Given the goals and objectives of the conference, we adopted principles from the Co-Design Framework and Inclusive Design Theory. Coined in 1988 by Forsgren, co-design is a framework that accounts for perspectives from multiple stakeholder groups and takes these into consideration by inviting feedback from diverse user groups [12]. Co-design shares many key principles with user-centered design and participatory design theories, in which stakeholders are invited to participate in all stages of the design process [13,14]. Similarly, we also considered principles of Inclusive Design Theory by wanting to create an end product and service that was accessible to as many people as possible without requiring adaptations or creating stigmatization [15,16]. Inclusive design also requires that a diverse range of end users is both considered and involved in the design process [15]. Together, these two theoretical frameworks provided clear guidance on how to engage our primary stakeholder groups during the conceptualizing, planning, and procurement phases of the conference. 

### 4.2. Preliminary Research

We adopted the International Classification of Function, Disability and Health (ICF) model of disability [17] to ensure that we considered accessibility from multiple lenses and considered barriers to participation as a limitation of the webinar technology and not because of a person’s disability or impairment. We also wanted to ensure that the presenters selected to speak at our conference represented a wide variety of impairments and disability types, such that the conference would include a range of perspectives. We also applied principles of a Co-design Framework [12] and Inclusive Design Theory [15] by engaging voices from the disability community throughout the entire planning, procurement, and delivery phases of the project. This was also in alignment with the United Nations’ CRPD Article 9, which states that appropriate measures should be taken to “promote the design, development, production and distribution of accessible information and communication technologies” at an early stage [8].

Our initial research was undertaken with three approaches:Surveying key stakeholders including individuals with disabilities and caregivers;Attending other accessible conferences to learn what platforms were being used and what accessibility features were being offered;Speaking to industry experts with a known focus on accessibility, including event coordinators, researchers, and service vendors.

This research helped inform decisions about the timing of the conference, conference themes, accessibility requirements, how to mitigate barriers to accessing the virtual event, and how to incorporate multi-sensory experiences. Based on this initial research, our team decided to incorporate the following elements in our conference:Closed captioning in both English and French;American Sign Language (ASL) interpretation;Langue des Signes Québécoise (LSQ, the form of sign language used by those fluent in French) interpretation;Audio interpretation between English and French;Hourly breaks to help offset the potential for “Zoom-fatigue” [18] and possible overstimulation.

For decisions relating to the timing and topics of the conference, as well as possible barriers not yet considered, feedback was solicited through a short online survey (via Survey Monkey). The survey was sent via email to a group of stakeholders, comprised of researchers who specialize in accessibility-related research and members of the disability community. Responses were collected over three weeks.

Many members of the research team signed up for other virtual conferences during this time, including those hosted by the Canadian Institute of the Blind, a11yTO, and International Centre for Evidence in Disability. This helped the team to develop an understanding of the accessibility services provided in virtual conference settings, what conference platforms were being used, and to network with event coordinators who prioritized accessibility. During these conferences, team members took notes related to the accessibility features and services and reported these back to the larger team for consideration.

Through our network of project stakeholders, research colleagues, and connections made at accessibility-related conferences, we began reaching out to experienced event coordinators and other researchers. The purpose of these meetings was three-fold: (1) to discuss our conference plan; (2) solicit feedback, and (3) seek recommendations for vendors, conference virtual platforms, and other conference-related tools. These meetings took place over Zoom and detailed notes were taken by the team members present, which were later disseminated to the rest of the team. A full scope of the work completed over the timeline of the conference planning can be seen in Table 1.

### 4.3. Results from Pre-Conference Survey

In total, 15 responses were collected from stakeholders across Canada, including researchers in fields related to accessibility and members of the disability community. Respondents were asked to respond to questions related to the timing, structure, and topics of the conference.

The results showed that nearly half of our respondents preferred an afternoon (Eastern Standard Time) conference (46.7%) taking place over three half days (78.6%). When asked what days of the week would be preferable for the conference, Wednesday (66.7%), Tuesday (60.0%), and Thursday (53.3%) were reported as the three most popular days. In an open-ended question, respondents were asked what specific topics they would like to hear about at the conference. The five responses to this question are displayed in Table 2. These results were then used to help set the dates, timing, and themes of the conference.

The decision to hold the conference over the course of three afternoons (based on Eastern Standard Time) came from the survey results in combination with a few other factors. Most of our team was based in Toronto, Canada, and given that the conference would be advertised across Canada with attendees and presenters spanning multiple time zones, holding it during Eastern Standard Time afternoon allowed relatively convenient attendance for anyone attending from within North America (attendees on the west coast would begin the conference at 9 a.m., and attendees on the east coast would begin the conference at 2:30 p.m.).

### 4.4. Decisions on Accessibility

Our team was committed to including closed captioning, ASL and LSQ sign language interpreters, spoken language interpreters (English–French), and breaks every hour to help reduce screen fatigue and possible overstimulation.

There were a number of comprehensive virtual conference platform services available that we considered. Many included auto-generated captions and had existing relationships with interpretive services. Alternatively, we could source our own services, including professional captioning services, which would have allowed us more control over the quality, expertise, and specific requirements we had for this conference. These decisions were at the forefront of many conversations held by the research team, and with industry experts and service providers.

The conference schedule was tentatively set prior to confirming any presenters. A decision was made to include a break in the conference programming for every 45 min of presentation time. This meant after every 45 min of presentations there was a 15-minute break. This was repeated for four hours each day. This break served two main purposes: (1) to allow the language interpreters and captioners a break from their work, and (2) to allow attendees to take a break from the screen. At one point, we considered streaming an audio-described video of hiking in a Canadian national park during the break, but ultimately concluded this may send conflicting messages and cause attendees to remain in front of their computers during the break time. In the end, we showed a static screen state that included website links where attendees could learn more about our research project.

Another step that the research team took towards ensuring the conference was accessible was deciding to work with presenters to help format and organize their presentations and materials. Traditionally, conference presentations often utilize slide decks that may include images, graphs, charts, and other visually descriptive features. If not described during the presentation by the presenters, these features may be inaccessible to attendees who are blind or visually impaired. To mitigate this, the research team developed an Accessible Presentation Guide that outlined guidelines and recommendations for how presenters could make their presentations more accessible. Presentation materials (e.g., presentation slides and transcripts) were requested in advance to send to the interpreters for their own preparation needs.

The research team also hosted optional drop-in sessions in the weeks leading up to the conference for attendees to get a sense of how to use all the features of our virtual platform and provide an opportunity for attendees to ask questions on what to expect during the conference.

A final component that helped improve the accessibility of the virtual conference was the decision to include a multi-sensory gift bag. This idea was conceived from a conversation with a researcher who had recently hosted their own accessible conference and included physical gift bags. These gift bags were developed with the primary goal of bringing a tactile and tangible aspect to what was otherwise an entirely virtual experience. The gift bags contained things that could be touched, smelt, and tasted by attendees, as well as highlighted accessible communications methods, such as Braille and ASL, and featured Canadian makers and artists.

## 5. Procurement

### 5.1. Selecting an Online Platform

Our team met with different virtual conference providers that offered comprehensive conference service packages that often included handling registration, presenter bios, agendas, and presenter materials, and supported virtual break-out rooms, multiple concurrent stages, and poster presentations. Some even allowed attendees to create avatars of themselves that would then be used to navigate within the virtual setting. The settings often included eye-catching graphics that created visually stimulating environments.

We consulted with stakeholders, and in particular individuals with disabilities, both in person and virtually, to learn what conference platforms they were familiar with and what ones worked or did not work for them based on their needs and experiences. While many providers claimed their services were accessible, we learned that each one was limited in some way or lacked one or more key features that we required. For instance, most were not able to support the number of on-screen and audio language interpreters we needed (e.g., a platform might support ASL and English captions, but could not support ASL, LSQ, captions in two languages, and language audio interpreters). Others had websites that seemed overly complicated with visual graphics that were incompatible with screen readers. Additionally, the more complicated the platform (e.g., requiring multiple clicks to locate information or navigate to each presentation), the more challenging it might be for individuals with limited dexterity or cognitive impairments to feel comfortable using the platform. Some platforms paired their services with other existing event-registration websites that our stakeholders told us were inaccessible to screen readers.

Ultimately, we decided to host the conference using Zoom Webinar (Zoom.us, at the time, considered the most accessible virtual conference platform) and develop our own website for housing registration, agenda, presenter biographies, and other materials. The Zoom Webinar platform also had the benefit of being widely used and was, therefore, most likely to be familiar to our attendees.

### 5.2. Website Development

Selecting the Zoom Webinar platform meant that it would be up to our team to create our own website. While it was more work, we felt it was worthwhile because it gave us more control over creating an accessible website. We had the freedom to keep the design simple, use high color contrast and large font sizes, and run our own tests for screen reader compatibility. Secondly, it gave us more flexibility overall, we could adjust and adapt the website as needed, adding new content as it was developed, and respond quickly to feedback we received about accessibility issues. Thirdly, it gave us a permanent place on which to house the conference materials, and as owners of the website we were not limited to a contract expiration date and could maintain the website for as long as we wanted.

We originally sourced a web-development company to build the website. We met with them to explain the goals of the conference, the importance of website accessibility, timelines, and what sort of ongoing maintenance would be required. The company assigned the project to a developer who began to design the website. Our team worked with the developer on multiple prototypes, trying to come to a satisfactory result. However, due to issues with communication, task execution, and understanding of our accessibility expectations, we dissolved the relationship at the end of the initial contract stage. We ultimately moved the development and maintenance of the website in-house to be managed by one of our team members (WS) who was familiar with web design. To meet the bilingual requirements of the conference, a French version of the website was created using a combination of French translation services and in-house translation by team members.

Future changes were iterative in nature, primarily based on feedback from individuals with disabilities and ongoing consultation with stakeholders. An example of one such change was an issue with webpage re-flow when users with vision impairments zoomed in on the website page. Due to technical limitations with the website platform, we were unable to resolve this issue on our primary website and instead created a secondary, large-font website with re-flow capabilities. The secondary webpage housed identical information but in a more accessible format for individuals with low vision.

### 5.3. Language and Interpretation Needs

This conference had an added level of complexity by being bilingual as well as taking into consideration accessible language requirements, such as ASL and descriptive language. Leaning on our network of stakeholders, including individuals with disabilities, industry experts, and researchers, our team began collecting names of organizations and businesses that supply interpreter and language services. We began interviewing these vendors after making the decision to host the webinar on Zoom and manage the vendor hiring and logistical organization ourselves.

After multiple conversations with vendors and considering the technical limitations of Zoom Webinar, we came to the decision to hire three vendors to supply the language and interpretation services we needed for the conference. The three vendors provided the following services: (1) ASL and LSQ interpretation, (2) live English and French captioners, and (3) French–English (audio) language interpreters. Including 15-minute breaks every hour allowed us to hire only two ASL interpreters and two LSQ interpreters, who alternated every 15 to 20 min. This break schedule helped reduce costs in relation to language and interpretation needs.

A benefit of using Zoom Webinar was that it supported bilingual audio at the conference. We employed a French interpreter to provide French audio for all of the English presentations, and an English interpreter to provide English audio for all of the French presentations. Zoom supports a bilingual audio feature that allows attendees to select which audio they wish to hear; the original or the interpretation. If the interpretation is selected the original audio is lowered to 20% volume and the audio interpretation is overlayed at 80% volume.

Further efforts to encourage bilingual participation in our conference included creating a French language version of the conference website, putting out a call for French presenters, and running advertisements in both English and French. Multiple members of our team were bilingual (MP-P, MG, AW) and, thus, we were able to use a combination of internal and external translation services to support the French language website and conference materials. 

### 5.4. Selecting an Event Producer

In the process of interviewing different interpretation and captioning vendors, it became clear that the magnitude of the conference was expanding beyond the capacity of our team. We decided to hire an event production company to look after much of the technical and coordination logistics. Once again leaning on our expanding network, we began reaching out to event producers with experience in the accessible conference sphere. 

It was important to our team that the event producer selected had experience producing accessible conferences and could understand the complexity of having both an accessible and bilingual conference. The producer we hired had experience producing events for the Canadian federal government and came with her own network of technicians, service vendors, and experience working with the Zoom Webinar platform. 

The research team was able to outsource the following tasks to the event producer: Rehearsals with all interpreter vendors and conference presenters;Informal dress rehearsal for the research team, who all played substantial roles in the implementation of the conference;Zoom Webinar screen layouts and Zoom Webinar casting from the Zoom Meeting (“green room”);Zoom Webinar recording and post-production editing;Day-of technicians who managed broadcasting from the Zoom Meeting to the webinar, screen transitions, and slide transitions;Day-of and last-minute technical changes to screen states and interpreter labels.

Hiring an event producer allowed our team to outsource some of the more complex technical aspects of hosting the conference and refocus our efforts on the other components, such as presenter preparation and creating an accessible gift bag. 

### 5.5. Presenter Selection and Preparation

The purpose of the conference was to gather people who share a common interest in making Canada’s national parks more accessible to individuals with disabilities. The primary goal was to ensure that the lived experiences of individuals with disabilities who face barriers in accessing national parks were represented. The call for presenters was open to anyone including people with lived experiences, caregivers, researchers, and organizations (both non-for-profit and for-profit) that help individuals with disabilities to access national parks and similar protected areas. The research team was committed to ensuring that a variety of voices would be represented at the conference and that the talks would not be limited to just one disability type. Our 40 presenters in total included people with mobility impairments, vision impairments, hearing impairments, sensory and cognitive impairments, researchers, and professionals working towards improving park accessibility. Presenters were assigned an individual time slot, a shared time slot, or were asked to sit on one of two panel discussions.

From our own experiences speaking at conferences and knowing that some of the presenters selected had limited public speaking experience, our team wanted to ensure that everyone felt supported and prepared to take on this task. We developed an Accessible Presentation Guide in collaboration with the event producer which included information about how the conference day would proceed, tips for presenting virtually, recommendations about lighting and camera placement, and guidance for creating an accessible presentation. The guide included suggestions, such as using large sans-serif fonts, using high color contrasts, and providing verbal descriptions of any images, charts, or tables. The document itself was shared in Word for screen reader compatibility and included alt-text for images. This document was shared with presenters shortly after they confirmed their speaking spot and before the deadline for submitting presentation materials. 

In addition to sharing the Accessible Presentation Guide, our team connected with presenters individually to answer any questions or concerns regarding their presentation. We also hosted optional meet-and-greet sessions for the panelists. The conference included two panel discussions: one on Day 2 focusing on barriers to accessing national parks, and one on Day 3 focusing on innovative solutions to park accessibility. The panels were comprised of four to five presenters and included individuals with disabilities, researchers, and members of organizations. The optional meet-and-greet sessions provided an opportunity for the panelists to meet each other in advance and to go over some of the questions that they would be asked as part of the panel. 

The final steps the team took to support presenters and mitigate any accessibility issues was requiring presenters to submit their slides in advance and attend a mandatory “tech check”. The requirement to submit slides in advance was for two purposes: (1) it allowed our team to review the slides for any accessibility concerns and work with the presenter to address them; and (2) the materials were then sent to the interpreters in advance of the conference, allowing them to familiarize themselves with the materials and any unique terminology. The tech checks were hosted by our event producer and helped ensure that each presenter had the necessary technology available to be seen and heard in the virtual conference. This also provided an opportunity to explain how the conference would be comprised of a Zoom Meeting (“green room”) and a Zoom Webinar (the conference).

### 5.6. Gift Bags

The idea for the event gift bags came from one of the conversations we had with an industry expert who had recently hosted an accessible conference and created their own physical gift bags to mail to attendees. The purpose of the event gift bags was two-fold: (1) to provide a tangible experience to what was otherwise a completely virtual event, and (2) to add a tactile and multi-sensory component to the accessibility of the conference. We also took this opportunity to reach out to Disabled artists and makers within Canada and featured their items in the gift bag. It was important to the team that the gift bags reflected the overall objective of our conference, being both accessible and highlighting the voices of individuals with disabilities within Canada. 

The gift bags were comprised of eight items, each carefully curated to align with the conference theme and mission:Braille card: a four-by-four inches card that was printed with three national park-related words in Braille and included the English words underneath;ASL card (with video): a four-by-six inches card that featured signing instructions for six national park-related words, as well as a QR code that linked to YouTube videos demonstrating the signing gestures;Tea bags: two tea bags from a Canadian tea manufacturer brought an olfactory and taste component to the gift bag;Art prints: we featured three different Disabled Canadian artists in the gift bag whose art was inspired by Canada’s natural beauty;Conference button: we worked with a Disabled artist to design a unique button for our conference;Parks Canada postcard: a four-by-six inches official Parks Canada postcard featuring one of 41 different Parks Canada sites;A brief write-up about the larger objective of our research project (in Braille and English text).

Our team printed and attached a Braille label to each physical item explaining what it was. The physical items were packaged into small envelopes and mailed to the first 250 attendees who provided a Canadian mailing address. 

As the conference attendee numbers grew beyond the supplies that we had, we came up with the idea of creating a virtual version of the gift bag. These also served the dual purpose of allowing us to provide longer, more detailed descriptions of each of the physical items for our visually impaired attendees. Members of our team who had previously taken a course on describing images for people who are blind or visually impaired utilized those learned skills to write descriptions of each of the physical items. We also included images of the items for the sighted attendees. The virtual gift bag was shared in a Microsoft Word document for its compatibility with screen reader technology and posted on our conference website.

## 6. Conference Delivery

### 6.1. Presentations

The three-day conference was delivered using a combination of online platforms consisting of a Zoom Meeting, Zoom Webinar, StreamText (streamtext.net), YouTube (youtube.com), WhatsApp (whatsapp.com), and OBS Studio (obsproject.com). The integration of these platforms for our conference delivery is shown in Figure 1.

The Zoom Meeting served as a “green room” and broadcasting studio. This is where the event producer and the technical team resided during the conference, as well as the conference emcee, the ASL/LSQ interpreters, and the presenters presenting each hour. The presenters for the upcoming hour would arrive in the Zoom Meeting approximately 10 min before the top of the hour. They would receive a debrief from the event producer and a quick audio and video check from the technician. Attendees of the Zoom Meeting would receive a count down and then the Zoom Meeting would be broadcasted to the Zoom Webinar and live-streamed to YouTube in both English and French. The emcee announced each presenter, and they presented their slides.

Zoom Webinar was used as the primary conference platform for attendees. Attendees were welcomed into the webinar at the beginning of each conference day and could remain in the webinar for the rest of the day. During the 15-min breaks, a static screen would be displayed to attendees in the webinar showing the time the presentations would resume. During each presentation, the conference attendees would see the presenter speaking as well as an ASL interpreter and LSQ interpreter on their screens, as shown in Figure 2. Attendees also had the option of switching between English and French audio. Unfortunately, while Zoom provided a lot of bilingual support, we were limited to providing captions in a single language within the Zoom Webinar itself. English captions provided by a live captioner were included in the Zoom Webinar and French captions were sent to a platform called StreamText. The link for French captions was provided in the webinar invite and Zoom Webinar chat feature during the conference.

Finally, we also created a pair of YouTube live streams of the Zoom Webinar using OBS Studio, which ran on a computer of one of our team members (ZG): one with English audio, English captions, and ASL; and the other with French audio, French captions, and LSQ. These live-streams allowed us to have more control over the screen state of the webinar. Within the Zoom Webinar, attendees were shown both the ASL and LSQ interpreters on their screen along with the presenters (see Figure 2). This proved to be overstimulating for some viewers and Zoom, at the time, did not allow attendees to customize their screen states. Live streaming also provided a recording of our event that was immediately available to anyone who wanted to watch the event asynchronously. Finally, we embedded both YouTube live-streams to our conference website to allow for a simple way for any conference attendees to attend. All of our conference videos can be viewed on our YouTube channel (YouTube.com/@EngineeringHealthResearch).

Due to the one-way communication between the Zoom Meeting and the Zoom Webinar, we had members of the research team in both rooms. Team members in the webinar were able to monitor for technical issues such as screen freezing or audio dropouts, and were able to speak directly to webinar attendees when needed (e.g., when technical difficulties occurred or when there were delays). Team members in the Zoom Webinar communicated with the team in the Zoom Meeting through a WhatsApp chat. All team members joined a short debrief meeting at the end of each day to discuss any issues that had come up or any last-minute changes to the agenda.

Finally, the chat feature in the Zoom Webinar was disabled because our testing showed that it conflicted with screen readers. The chat feature was temporarily enabled in instances when our team needed to communicate information to attendees such as delays in the conference’s start time, links to access French captions, or the methods for submitting questions.

### 6.2. Taking Questions

Our set-up presented a challenge for taking questions from conference attendees because the presenters/panelists in the Zoom Meeting could not see or hear the attendees in the Zoom Webinar. We addressed this problem by enabling the Question and Answer box in the Zoom Webinar and by providing a dedicated email address and phone number for the audience to pose questions, as shown in Figure 3. Members of our team collected the incoming questions and relayed the question to the emcee in the Zoom Meeting using the website service Slido (slido.com).

### 6.3. Costs

Thanks to the funding provided by Accessibility Standards Canada, our team was able to make the conference free for all attendees without the need to attract sponsors. In Canada, individuals with disabilities face higher rates of unemployment [19], often living on fixed incomes, making cost a potential barrier to participation. We did not want cost to prohibit anyone from attending this conference and engaging in conversations on improving access to national parks. 

The total conference cost was approximately $67,000 CAD; a high-level breakdown of costs is provided in Table 3. Our team members’ salaries are not included in this table, but their roles and responsibilities have been outlined in Table 4. 

## 7. Conference Evaluation

### 7.1. Attendance

In total, 606 people registered for the conference from across Canada and internationally. On average, from 150 to 200 people attended the conference each day, with the exact number fluctuating throughout the day, and, as of this writing, we have had a combined total of 388 views of the recorded conference videos posted on YouTube [20].

### 7.2. Post-Conference Survey

Conference attendees were sent a short post-conference survey, designed with Qualtrics, which asked some demographics-related questions, whether the respondent felt their accessibility needs had been met, what aspects of the conference were done well, and what could be improved on for next year’s conference. Attendees were sent the survey via email and responses were collected over three weeks. We switched to Qualtrics (from Survey Monkey, which was used for the pre-conference survey) for the post-conference survey after learning that Qualtrics provides many more accessibility features including large-font text, high-contrast colors, and customization for screen reader technology compatibility. 

In total, 26 responses were collected from attendees. Participant demographics are summarized in Table 5.

When asked if they were satisfied with the way the conference team handled their accessibility needs, 16 attendees responded yes, 4 responded no, and 6 did not respond to this question. The four attendees who responded no explained that they did not have any accessibility needs for the team to address (see Table 6).

The final questions of the post-conference survey asked attendees what they thought of the conference and solicited ideas to improve the conference in the future. Of the 26 survey responses received, 24 attendees responded to this question. The feedback included general appreciation for the conference, with many respondents stating that they enjoyed the conference and felt it was valuable and informative:


*
Overall I really thought it was great and I took away some really good info. I will be encouraging management at my park to watch the sessions.
*


*The conference was great. I appreciated the range of presentations; some academic, some organizational, some informational and some experiential*.

Recommendations for improvement were mostly related to three ideas: (1) longer presentations and longer question periods; (2) improved networking opportunities; and (3) more focus on actionable solutions and resources for improving park access. 

In general, attendees found the 15-minute-long presenter timeslots to be too short, and they commented that the presentations “felt rushed” and did not allow enough time for questions:


*
Allow more time overall. Each presentation felt rushed and not enough time for questions. Allow more time per presenter so that people feel open to ask questions.
*



*
I personally would have liked slightly longer sessions for some of the speaker. 15 min wasn’t really enough time to get into the meat of the lectures.
*


Attendees also commented that they would like to see more opportunities for networking and open discussion with presenters. The one-way communication feature of the Zoom Meeting and Zoom Webinar limited the networking opportunities of the conference, and it was apparent from the feedback that attendees would like to see that changed in the future:


*
It would be neat to have a space to connect with accessibility experts and have the chance to have open communication with them throughout the conference.
*



*
I would have preferred more time to meet, connect and discuss, as well as participate.
*


While many attendees enjoyed the breadth of presentations and topics discussed, some of the feedback highlighted the value of hearing about actionable solutions for improving access at parks and existing resources that land managers can utilize now. With many land managers at both national and provincial park levels in attendance, it makes sense that they are seeking solutions and actionable steps they can take away and implement immediately to help improve access at their parks:


*
The sessions I found the most interesting were those that focused on best practices and solutions to reduce and remove barriers. Things that we can implement or work towards.
*



*
For the next conference it would be great to learn about more resources or toolkits people can access to support their work in increasing access to parks and trails.
*


### 7.3. Reflections from Our Team

After the conference ended, our team held a series of debriefing sessions. These served as an opportunity for the team to reflect on the aspects of the conference that went well and where there were areas for improvement. 

Overall, our team felt that the conference went well. We felt that we were able to achieve our goals of bringing together people with a shared desire to improve park accessibility and creating an accessible, bilingual conference. Some strengths included being able to pivot quickly on the first day despite initial technical difficulties and incorporating suggested screen state layouts from attendees. This change enabled us to offer a second screen state option for the conference attendees through the YouTube stream that reduced the number of onscreen interpreters, thus reducing the onscreen stimulation. Another strength was the decision to have a single dedicated conference emcee and panel moderator throughout the conference. This allowed for one consistent and familiar voice for attendees who were blind or visually impaired during a virtual conference that included 40 presenters and, thus, 40 unique voices. There was also a consensus among the team, and from the emailed feedback, that the physical gift bags were well received and created a positive hands-on experience during the virtual event. Finally, the team agreed that a decision to create a single landing page on our website that included all the relevant website links discussed during the conference provided a helpful resource for attendees and limited the number of places we needed to direct attendees to for information. 

Our team also came up with a number of ways that a future conference could be improved. First, this included creating a dedicated email address for all conference-related communications, particularly communication with presenters. In the planning stages, we had multiple team members communicating with presenters and this led to confusion and missed communication for both our team and the presenters. Another opportunity for improvement was presenter training. While we provided the Accessible Presentation Guide with guidelines and recommendations for creating accessible slides, as well as one-on-one support for developing presentation materials, there were still some concerns with the accessibility of some presentations and pacing (it can be difficult for interpreters and captioners to keep up if presenters move too fast). Comprehensive and mandatory presenter training, providing presenters with a slide deck template, and allowing the research team additional time to review presentations in advance and work on an individual level with the presenters that require additional support, would help mitigate this concern in the future. A final area for improvement discussed was opportunities for advertising. We used multiple advertising streams including social media, paid Google ads, podcast advertisements, online magazine advertisements, and our professional research and industry networks; however, we still faced challenges reaching land managers. Beginning advertising earlier and creating more targeted advertisements and outreach to land management organizations could improve the attendance rate in that area.

### 7.4. Implications for Universal Design and Broader Applications

Many industries that have been historically exclusionary continue to struggle with how to implement universal design and how to appropriately account for the different needs within the disability communities. Over the last four decades, universal design has gained popularity and is being increasingly adopted in the design of physical spaces. However, this work highlights the importance of expanding our conception of universal design to include all interactions that internet communication technologies provide. The latest textbook on universal design contains only a single page out of nearly 400 pages on internet communication and highlights the need for a societal shift to include the same level of detail in our recommendations of virtual spaces as we provide for physical spaces [21]. This work helps address some of those gaps within universal design of technology and technology-supported communication. 

In particular, there is a movement towards creating “smart cities”, a multi-faceted term that ultimately strives for the “social inclusion of all urban residents” [22]. This movement demonstrates the growing awareness of the importance of accessibility and inclusion in both physical and virtual environments. Vincent Cerf, the inventor of commercial email, lived with a hearing impairment and was married to a woman who was deaf. He noted “because so much of the communication on the Internet is in written form…people with hearing loss are often placed on completely equal status with those of normal hearing” [23]. Well-designed virtual meetings that include audio, video, sign language, captioning, audio descriptions, and simple user-interfaces, and are screen reader compatible have the potential to be equitable to many more individuals and to deliver even greater direct and indirect benefits to everyone in our society.

## 8. Recommendations for Future Conferences

Our team learned many lessons over the course of the conference planning and delivery. We have the following recommendations for others planning similar conferences in the future: **Include individuals with disabilities from the start:** Our team put together a diverse group of stakeholders who supported our larger project work and played an invaluable role in the planning of this conference. They provided endless and ongoing feedback for accessibility needs. We also asked about the accessibility needs of attendees during the conference registration phase and encouraged attendees to get in touch with our team directly. This allowed us to better understand potential barriers in a virtual conference and how to address them.**Understand the scope of work covered:** When working with vendors, particularly those that utilize an hourly fee structure, it is important to understand and communicate the full scope of work that is anticipated. This includes preparation time, delivery of the service or product, and any post-conference production, such as re-recording segments or post-production editing.**Understand cost structures:** Many of the language interpretation services were billed at an hourly rate with a set minimum number of hours for shorter projects. Additionally, sign language interpreters often work in teams of two or more so that they can alternate signing and take required rest periods. For conferences that span multiple hours there may be a need for a larger team of interpreters.**Plan to work with presenters to improve presentation accessibility:** Our conference included a mix of presenters from many backgrounds, and, as such, many had not spoken at a conference before. We wanted our presenters to feel supported and informed, particularly with regards to preparing an accessible presentation. The Accessible Presentation Guide helped, but there were still challenges when it came to communicating what some of the barriers are to visual presentations. We learned that many presenters would have benefited from having slide templates and mandatory accessibility training sessions. Alternatively, consider challenging presenters to present without the use of slides.**Importance of rehearsals:** Our event producer hosted mandatory rehearsals for each of the presenters, as well as for our team. These were invaluable in ensuring that presenters had the necessary technology (e.g., internet, computer, camera, microphone) and a comfortable environment from which to present (e.g., good lighting, minimal distractions in the background). It also allowed the research team an opportunity to meet with the presenters and help them become comfortable with the virtual platform.**The more time to plan, the better:** While planning for the conference got underway in January of 2022, eight months prior to the conference, we did not begin advertising the conference or soliciting presenter applications until May. This limited the amount of time our team could spend reviewing applications, selecting presenters, and working with them to prepare accessible presentations. It also limited the amount of advertising we could do for the conference, and potentially limited our attendance.**Website accessibility:** Achieving good website accessibility can be tricky. While our team put considerable effort into ensuring our conference website included large fonts, high-contrast colors, image descriptions, and simple user interfaces, we failed to check how the elements on the site would shift if someone chose to zoom-in on a portion of the screen. We learned the platform our website was hosted on did not support website content re-flow, thus making the website inaccessible to visitors with low vision. Future conferences should check whether the website platform supports content re-flow prior to website development.**Accessible networking opportunities:** We ended up not having any networking opportunities because we were unable to supply enough interpreters/captioners to be present in multiple breakout rooms. Future consideration should be given to providing meaningful and accessible networking opportunities at virtual conferences.

As the digital world continues to expand and our engagement and socialization in online settings broadens, it is important that event planners and conference organizers consider how they can maximize the accessibility of their events so that as many people as possible can participate. The popularity of virtual events is growing and the opportunities for inclusion are expanding. Many of the platform vendors we spoke with acknowledged their accessibility shortcomings and were working to improve their systems.

We expect the growing awareness of the importance of accessibility and inclusion will lead to continuous improvement in this field and make the process of putting on accessible, bilingual, virtual conferences easier in the future. Our hope is that this case study can serve as a road map for event planners and conference organizers, who may be new to the space of accessible design and assist them in creating events that are both accessible and bilingual, meeting the needs of all event participants and attendees and, thus, ensuring equitable inclusion. 

## Figures and Tables

**Figure 1 ijerph-20-02302-f001:**
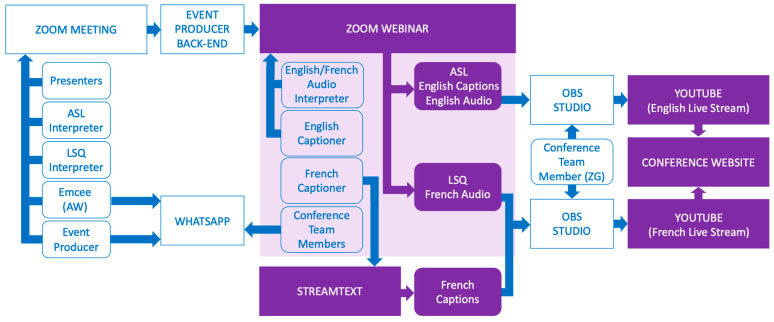
This diagram shows the platforms (Zoom Meeting, Zoom Webinar, StreamText, YouTube, WhatsApp, and OBS Studio), individuals, and interpretation/accessibility features that were needed to deliver our conference and how they were connected. The platforms and elements that the audience was able to access are indicated in purple.

**Figure 2 ijerph-20-02302-f002:**
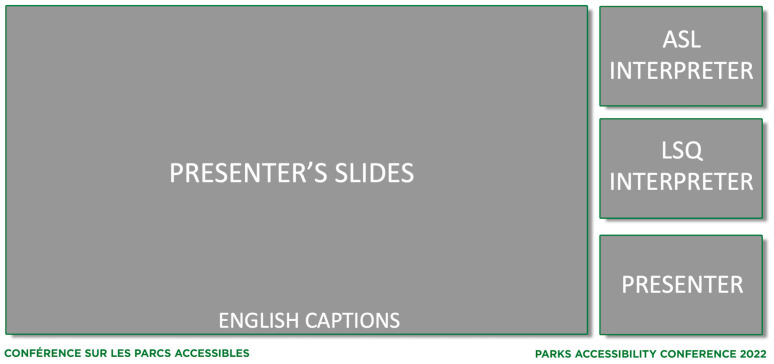
Screen state of the Zoom Webinar for single presenter with presentation slides.

**Figure 3 ijerph-20-02302-f003:**
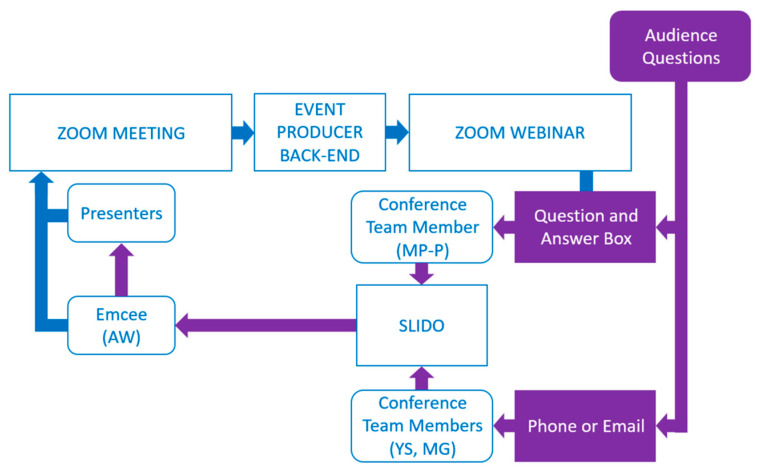
This diagram shows our approach to receiving and having audience questions answered by the presenters.

**Table 1 ijerph-20-02302-t001:** Conference timeline from January to September 2022.

Month	Task
January	Looked for accessible conference platforms, decided on conference timeline and virtual platform, launched pre-conference survey.
February	Continued researching accessible conference platforms, interviewed potential virtual conference platforms, drafted presenter application, drafted advertising materials.
March	Continued researching accessible conference platforms, interviewed potential virtual conference platforms, built draft website for the conference.
April	Hired website designer to create an accessible conference website, began advertising on social media and through partner organizations, met with potential vendors for interpretive needs.
May	Began confirming keynote presenters, partnered with Canadian vendors for physical gift bag items, launched registration and presenter applications on conference website, expanded advertising efforts to podcasts and magazines, met with potential vendors for interpretive needs.
June	Continued advertising, hired a production coordinator, confirmed vendors for interpretive needs, and confirmed vendors for physical gift bag.
July	Continued advertising, closed presenter applications, assessed presenter applications and selected presenters, mapped out daily agendas, contacted presenters to request materials, and created a digital gift bag.
August	Continued advertising, collected presenter materials and conducted accessibility checks and improvements where possible, held technology tests with presenters, held dress rehearsal with tech team, packed and shipped gift bags, held conference.
September	Post-conference tasks including post-conference feedback survey, transcript editing, video editing.

**Table 2 ijerph-20-02302-t002:** Pre-conference survey responses.

Respondent	Response
1	“Accessible trails and photo opportunities, accessible parking, accessible benches, accessible water fountains, accessible doors, accessible transportation, accessible accommodations: camping, hotels, accessible residencies, activities, workshops and tours.”
4	“How parks can be used to promote physical activity and wellness for people with disability, and policies related to this.”
6	“Accessibility planning requirements under new legislation how organizations will approach this work.”
10	“Wayfinding, access to information, navigable trails, amenities.”
12	“Inclusive programming.”

**Table 3 ijerph-20-02302-t003:** Conference cost breakdown.

Service/Product	Cost
Language and interpretation services (ASL, LSQ, captions, audio translations)	$22,000
Event production	$17,000
Gift bags (products, services, mailing costs)	$8500
Advertising and marketing	$5000
Presenter honorariums	$4100
Website hosting	$3500
Website accessibility	$3400
Zoom Webinar	$3300
Miscellaneous supplies	$200
Total	$67,000

**Table 4 ijerph-20-02302-t004:** Roles and responsibilities of research team members.

Team Member	Role and Responsibilities
YS	Lead Conference Coordinator. Led conference planning including sourcing, interviewing, and selecting webinar platform; interpretation services and event production team. Led conference website design. Designed registration and presenter applications. Reviewed presenter applications and selected presenters. Coordinated advertising. Developed Accessible Presentation Guide for presenters in conjunction with event producer. Held attendee drop-in sessions. Oversaw packing and shipping of physical gift bags. Attended technical checks with event producer. Monitored phone line during the conference. Provided leadership and ensured deadlines were met.
MG	Co-lead Conference Coordinator. Led conference planning including sourcing, interviewing, and selecting interpretation services and event production team. Designed registration and presenter applications. Reviewed presenter applications and selected presenters. Coordinated advertising. Developed Accessible Presentation Guide for presenters in conjunction with event producer. Held attendee drop-in sessions. Sourced goodie-bag vendors and items. Developed digital gift bag. Attended technical checks with event producer. Monitored email during conference.
ZG	Technical Support. Provided technical expertise when interviewing and selecting webinar platform, interpretation services, and event production team. Reviewed presenter applications and selected presenters. Developed introduction to Zoom platform video for attendees. Provided back-up technical support for conference and managed YouTube streaming of conference. Recorded conference and provided post-production editing of conference recordings for YouTube.
WS	Technical Support. Provided technical expertise when interviewing and selecting webinar platform, interpretation services, and event production team. Reviewed presenter applications and selected presenters. Assisted with website development and ongoing maintenance. Provided back-up technical support for conference. Primary contact with AW during conference.
AW	Conference Support and Conference Emcee. Assisted with conference planning, hiring of interpretation services, and event production team. Reviewed presenter applications and selected presenters. Assisted with in-house translations. Assisted with advertising. Organized and issued presenter honorariums. Emceed the conference and moderated the panels.
MP-P	Conference Support and In-House French translation. Assisted with conference planning, hiring of interpretation services, and event production team. Reviewed presenter applications and selected presenters. Handled in-house French translations. Assisted with advertising. Monitored the Zoom Webinar Q&A Box during conference.
ND	Conference Support. Assisted with conference planning, hiring of interpretation services, and event production team. Reviewed presenter applications and selected presenters. Assisted with advertising.
HG, HK, SA	Conference Support. Assisted with the packing and shipping and physical gift bags.
TD	Principle Investigator. Overall supervision and management, budget management. Assisted with advertising. Presenter at conference.

**Table 5 ijerph-20-02302-t005:** Respondent demographics.

Demographics	N (%)
Land management organizations	10 (38.5%)
Individual with a disability (i.e., pain-related disabilities, dexterity, mobility, mental health, seeing/vision, hearing, learning, memory, development, and Deaf/Blind)	7 (26.9%)
Organization supporting individuals with disabilities	5 (19.2%)
Caregiver	2 (7.69%)
Student	2 (7.69%)
Researcher	1 (3.85%)
Other	7 (26.9%)

**Table 6 ijerph-20-02302-t006:** Responses to the question “Were you satisfied with the way the conference team handled your accessibility needs? —No, please specify”.

Respondent	Response
2	I didn’t have any requests
23	Not applicable
24	I attended via Zoom—from my home! I did not have to go anywhere and I had access to the things that [made] my life easier.
25	I didn’t have accessibility needs but was impressed with the extent of accessibility provided by the conference.

## Data Availability

Not applicable.

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
