# Peer review of "Virtual Accessible Bilingual Conference Planning: The Parks Accessibility Conference"

_ijerph, 2023, doi:10.3390/ijerph20032302_

Round 1
Reviewer 1 Report
Thank you for the opportunity to review the manuscript ‘Virtual Accessible Bilingual Conference Planning: The Parks 2 Accessibility Conference’. The manuscript is a case study of a co-design and co-production process to ensure virtual conferences are inclusive of the needs of people with disability. This approach is important given the increase in the use of virtual conferences in an era of a Covid-19 pandemic, hence the relevance of this manuscript to other conference organisers.
The details of the co-design and co-production approach in the manuscript provides a practical direction of how other conference organisers can adopt to ensure disability inclusion. As such, there is good utility and implication for other conference organisers to learn and adopt the approach. It is a well written manuscript and it is easy to follow in terms of implication.
Whilst I found the manuscript practical and useful, it is important to note that the process is not unique and in some sense does not add to new research evidence per se. And as such, I have referred this manuscript as a case study or simply a ‘show and tell’ of an approach. For those of us who work mainly in the disability sector, such co-design process is a typical or standard requirement.
I note the manuscript refers to a universal design theory but contemporary practice refers to a co-design and co-production which are social or public policy approaches. The manuscript could consider references to the United Nations Convention on the Rights of Persons with Disabilities (2006) with the following relevant Article 5 (Equality and non-discrimination), Article 9 (Accessibility) and Article 21 (Freedom of expression and opinion, and access to information) as the most relevant Articles.
Author Response
We would like to thank the editor and the reviewers for their time in reviewing our paper and providing kind and thoughtful feedback. We are grateful for the opportunity to revise this paper. We have addressed all criticisms and suggestions below. The thoughtful recommendations have improved this paper considerably and have made it more likely that this paper will be of value to conference coordinators and event planners wishing to make their virtual events more accessible to people with disabilities.
Reviewer 1: Whilst I found the manuscript practical and useful, it is important to note that the process is not unique and in some sense does not add to new research evidence per se. And as such, I have referred this manuscript as a case study or simply a ‘show and tell’ of an approach. For those of us who work mainly in the disability sector, such co-design process is a typical or standard requirement.
We would like to thank Reviewer 1 for sharing this perspective and agree that we our intention with this paper was not to introduce a new design process but indeed to provide a case study to share the practical details that went into designing and producing a bilingual, virtual and accessible conference. We have added some text to the sections 2.0 (Objective) and 8.0 (Conclusion) to clarify:
Section 2.0: The objective of this paper is to share the approach we took, along with the key lessons we learned in planning and delivering our Parks Accessibility Conference in the summer of 2022, as a case study.
Section 8.0: Our hope is that this case study can serve as a road map for event planners and conference organizers, who may be new to the space of accessible design and assist them in creating events that are both accessible and bilingual, meeting the needs of all event participants and attendees and thus ensuring equitable inclusion.
Reviewer 1 noted: I note the manuscript refers to a universal design theory but contemporary practice refers to a co-design and co-production which are social or public policy approaches. The manuscript could consider references to the United Nations Convention on the Rights of Persons with Disabilities (2006) with the following relevant Article 5 (Equality and non-discrimination), Article 9 (Accessibility) and Article 21 (Freedom of expression and opinion, and access to information) as the most relevant Articles.
We are very grateful for you sharing your expertise and for providing us an opportunity to review our manuscript. Upon review we realised we had intended to reference Inclusive Design Theory, and not Universal Design Theory. Inclusive Design Theory shares many of the same principles of a Co-design Framework and together these two theoretical frameworks much more accurate reflect the methodological approach we took to planning and implementing the conference. We are also grateful for the suggestion to review the United Nation’s Convention on the Rights of Persons with Disabilities and have added references to articles 9 and 21 as a result of this recommendation in section 1.0:
The United Nations’ Convention on the Rights of Persons with Disabilities (CRPD) provides broad guidelines on access and equity issues for persons with disabilities, including information pertaining to accessing information. The CRPD, Article 21, notes that persons with disabilities have the right to seek information “through all forms of communication of their choice”, including accessible technology, availability of sign language, augmentative and alternative communication [8]. Conferences are a well-known venue for knowledge and information sharing and seeking, both in person and virtually using online technologies.
We have added a new section 4.1 that explains the theoretical frameworks: Co-design Framework and Inclusive Design Theory
We have also added the following to section 4.2:
This is also in alignment with the United Nations’ CRPD Article 9, which states that appropriate measures should be taken to “promote the design, development, production and distribution of accessible information and communication technologies” at an early stage [8].
We have also reviewed the paper in its entirety and made corrections to grammar and spelling as required.
Kind regards,
Alison Whiting
Alison.whiting@uhn.ca
Tilak Dutta
Tilak.dutta@uhn.ca
Reviewer 2 Report
The paper is undoubtedly a very well structured and clearly expressed work. The document is a very interesting work as an example of how to plan an event from the perspective of universal accessibility.
It would be interesting to include a discussion on the contribution it makes in relation to the existing literature on the topic of “Universal Design” and “Smart Cities” which is broad and very diverse.
1. What is the main question addressed by the research?
The paper describes the procedure followed to plan a conference in an inclusive way, taking into account the needs of people with disabilities. To this end, the paper describes how people with disabilities are included in decision-making, the selection of virtual conference platforms, subtitlers and interpreters in what the authors describe as "a multisensory experience". The document also summarizes the results extracted from an evaluation survey of the conference attendees and the reflections of the work team on the positive aspects of the conference and the opportunities for improvement. As an outstanding contribution, it concludes with a set of practical recommendations for planning accessible virtual bilingual conferences.
2. Do you consider the topic original or relevant in the field? Does it address a specific gap in the field?
The paper is undoubtedly a very well structured and clearly expressed work. The document is a very interesting piece of work as an example of how to plan an event from the perspective of universal accessibility.
The contribution is interesting as procedure but it is not contributing to improving the scientific knowledge in the field. Its main contribution is to put into practice a set of actions aimed at making a public event for people with disabilities more inclusive and accessible.
3. What does it add to the subject area compared with other published material?
Practical procedure. Its main contribution is to put into practice a set of actions aimed at making a public event for people with disabilities more inclusive and accessible.
4. What specific improvements should the authors consider regarding the
methodology? What further controls should be considered?
The methodology is correctly presented, but it is not very relevant. It will necessary to include a specific section in which a discussion is carried out on the contribution of this work in relation to the existing literature on the topic
5. Are the conclusions consistent with the evidence and arguments presented and do they address the main question posed?
Yes
6. Are the references appropriate?
The references are not enough on the topic of "Universal Design" and "Smart Cities" which is wide and diverse but poorly referred to or neither discussed in the paper.
7. Please include any additional comments on the tables and figures.
Tables are right
Author Response
We would like to thank the editor and the reviewers for their time in reviewing our paper and providing kind and thoughtful feedback. We are grateful for the opportunity to revise this paper. We have addressed all criticisms and suggestions below. The thoughtful recommendations have improved this paper considerably and have made it more likely that this paper will be of value to conference coordinators and event planners wishing to make their virtual events more accessible to people with disabilities.
Reviewer 2 noted: The paper is undoubtedly a very well structured and clearly expressed work. The document is a very interesting work as an example of how to plan an event from the perspective of universal accessibility. It would be interesting to include a discussion on the contribution it makes in relation to the existing literature on the topic of “Universal Design” and “Smart Cities” which is broad and very diverse.
Thank you for your kind assessment of our work. We appreciate your comments on Universal Design and Smart Cities and agree that brings an interesting perspective to our paper. We have added the following text to our discussion in section 7.4:
Many industries that have been historically exclusionary continue to struggle with how to implement universal design and how to appropriately account for the different needs within the disability communities. Over the last four decades, Universal Design has gained popularity and is being increasingly adopted in the design of physical spaces. However, this work highlights the importance of expanding our conception of Universal Design to include all interactions that internet communication technologies provide. The latest textbook on Universal Design contains only a single page of nearly 400 pages on internet communication, and highlights the need for a societal shift to include the same level of detail in our recommendations of virtual spaces as we provide for physical spaces [21].
In particular, there is a movement towards creating “smart cities”, a multi-faceted term that ultimately strives for, “social inclusion of all urban residents” [22]. This movement demonstrates the growing awareness of the importance of accessibility and inclusion in both physical and virtual environments. Vincent Cerf, the inventor of commercial email, lived with a hearing impairment and was married to a woman who was deaf. He noted “because so much of the communication on the Internet is in written form…people with hearing loss are often placed on completely equal status with those of normal hearing” [23]. Well-designed virtual meetings that include audio, video, sign language, captioning, audio descriptions, simple user-interfaces and are screen-reader compatible have the potential to be equitable to many more individuals and to deliver even greater direct and indirect benefits to everyone in our society.
We have also reviewed the paper in its entirety and made corrections to grammar and spelling as required.
Kind regards,
Alison Whiting
Alison.whiting@uhn.ca
Tilak Dutta
Tilak.dutta@uhn.ca
Reviewer 3 Report
Thanks very much for this very interesting article. The account of the work carried out is very useful in filling a gap regarding how to make online conferences more accessible.
Author Response
We would like to thank the editor and the reviewers for their time in reviewing our paper and providing kind and thoughtful feedback. We are grateful for the opportunity to revise this paper. We have addressed all criticisms and suggestions below. The thoughtful recommendations have improved this paper considerably and have made it more likely that this paper will be of value to conference coordinators and event planners wishing to make their virtual events more accessible to people with disabilities.
Reviewer 3: Thanks very much for this very interesting article. The account of the work carried out is very useful in filling a gap regarding how to make online conferences more accessible.
Thank you for your kind words and support of this work.
We have also reviewed the paper in its entirety and made corrections to grammar and spelling as required.
Kind regards,
Alison Whiting
Alison.whiting@uhn.ca
Tilak Dutta
Tilak.dutta@uhn.ca
Reviewer 4 Report
The case presented is of great interest; however, it is necessary to introduce a qualified theoretical-methodological path. It would be useful for them to outline the theoretical approach to this experience. So presented could be a good case study well described. It is suggested to implement a theoretical framework.
Author Response
We would like to thank the editor and the reviewers for their time in reviewing our paper and providing kind and thoughtful feedback. We are grateful for the opportunity to revise this paper. We have addressed all criticisms and suggestions below. The thoughtful recommendations have improved this paper considerably and have made it more likely that this paper will be of value to conference coordinators and event planners wishing to make their virtual events more accessible to people with disabilities.
Reviewer 4: The case presented is of great interest; however, it is necessary to introduce a qualified theoretical-methodological path. It would be useful for them to outline the theoretical approach to this experience. So presented could be a good case study well described. It is suggested to implement a theoretical framework.
Thank you for this feedback and for providing us an opportunity to review our methodology and provide more information around the theoretical-methodological path we took. We have added the following new section 4.1 to clarify the use of Inclusive Design Theory and Co-design frameworks as part of the approach we took to planning and implementing our conference:
Given the goals and objectives of the conference, we adopted principles from the Co-Design Framework and Inclusive Design Theory. Coined in 1988 by Forsgren, co-design is a framework that accounts for perspectives from multiple stakeholder groups, and takes these into consideration by inviting feedback from diverse user groups [12]. Co-design shares many key principles with user-centered design and participatory design theories, by which stakeholders are invited to participate in all stages of the design process [13, 14]. Similarly, we also considered principles of Inclusive Design Theory by wanting to create an end product and service that was accessible to as many people as possible without requiring adaptations or creating stigmatisation [15, 16]. Inclusive design also requires that a diverse range of end users is both considered and involved in the design process [15]. Together, these two theoretical frameworks provided clear guidance on how to engage our primary stakeholder groups during the conceptualising, planning and procurement phases of the conference.
We have also reviewed the paper in its entirety and made corrections to grammar and spelling as required.
Kind regards,
Alison Whiting
Alison.whiting@uhn.ca
Tilak Dutta
Tilak.dutta@uhn.ca
Round 2
Reviewer 2 Report
Dear authors,Dear authors,
thank you for your feedback on your work. The modifications made considerably improve the quality of the paper.
Reviewer 4 Report
The proposed revisions were welcomed and implemented within the proposed framework. The work is interesting